# Effects of Gait Rehabilitation Robot Combined with Electrical Stimulation on Spinal Cord Injury Patients’ Blood Pressure

**DOI:** 10.3390/s25030984

**Published:** 2025-02-06

**Authors:** Takahiro Sato, Ryota Kimura, Yuji Kasukawa, Daisuke Kudo, Kazutoshi Hatakeyama, Motoyuki Watanabe, Yusuke Takahashi, Kazuki Okura, Tomohiro Suda, Daido Miyamoto, Takehiro Iwami, Naohisa Miyakoshi

**Affiliations:** 1Department of Orthopedic Surgery, Akita University Graduate School of Medicine, 1-1-1 Hondo, Akita 010-8543, Japan; twrofwsh0930@outlook.com (T.S.); miyakosh@doc.med.akita-u.ac.jp (N.M.); 2Division of Rehabilitation Medicine, Akita University Hospital, 44-2, Hiroomote Hasunuma, Akita 010-8543, Japan; kasukawa@doc.med.akita-u.ac.jp (Y.K.); dkudo@doc.med.akita-u.ac.jp (D.K.); hata@hos.akita-u.ac.jp (K.H.); m.w@hos.akita-u.ac.jp (M.W.); takayu@hos.akita-u.ac.jp (Y.T.); okura@hos.akita-u.ac.jp (K.O.); suda_pt@hos.akita-u.ac.jp (T.S.); daido@hos.akita-u.ac.jp (D.M.); 3Department of Systems Design Engineering, Faculty of Engineering Science, Akita University Graduate School of Engineering Science, 1-1 Tegata Gakuen-cho, Akita 010-8502, Japan; iwami@gipc.akita-u.ac.jp

**Keywords:** robotic rehabilitation, orthostatic hypotension, earlobe blood flow

## Abstract

Background: Orthostatic hypotension can occur during acute spinal cord injury (SCI) and subsequently persist. We investigated whether a gait rehabilitation robot combined with functional electrical stimulation (FES) stabilizes hemodynamics during orthostatic stress in SCI. Methods: Six intermediate-phase SCI patients (five males and one female; mean age: 49.5 years; four with quadriplegia and two with paraplegia) participated. The participants underwent robotic training (RT), with a gait rehabilitation robot combined with FES, and tilt table training (TT). Hemodynamics were monitored using a laser Doppler flowmeter for the earlobe blood flow (EBF) and non-invasive blood pressure measurements. The EBF over time and the resting and exercise blood pressures were compared between each session. Adverse events were also evaluated. Results: The EBF change decreased in TT but increased in RT at the 0.5-min slope (*p* = 0.03). Similarly, the pulse rate change increased in TT but decreased in RT at the 1-min slope (*p* = 0.03). Systolic and mean blood pressures were slightly higher in RT than in TT but not significantly (*p* = 0.35; 0.40). No adverse events occurred in RT, but two TT sessions were incomplete due to dizziness. Conclusions: RT with FES can reduce symptoms during orthostatic stress in intermediate-phase SCI. Future studies require a larger number of cases to generalize this study.

## 1. Introduction

Spinal cord injury (SCI) is a condition in which the neurons of the spinal cord are damaged, often due to events such as a fall or traffic accident. SCI results in axonal regeneration and synaptic depression, leading to various disorders, including motor impairment [1]. In Japan, 4603 traumatic SCI cases occur annually, based on a 2018 study [2]. The number of incomplete SCI cases among elderly people due to low-energy trauma (such as falls on level surfaces) is increasing. Given our aging society, the incidence of incomplete SCI will continue to increase. Prolonged activity restriction can lead to complications, such as pneumonia, severe decubitus ulcers, and urinary tract infections, which are sometimes life-threatening [2]. Elderly patients are fragile and can easily develop muscle atrophy. For example, after only 10 days of bed rest, 10–14% of lower-extremity muscle strength is lost in the elderly [3]. Moreover, SCI patients are prone to synergistic muscle atrophy [4]. Therefore, early intervention with rehabilitation for SCI is important [5].

Orthostatic hypotension (OH) is a common complication of SCI, occurring in 50% of people with paraplegia and 82% of those with tetraplegia immediately after SCI [6]. OH causes dizziness and lightheadedness. It interferes with the rehabilitation of SCI patients and significantly impairs their quality of life [7]. Therefore, more patients, regardless of age, are forced to stay in bed for longer periods. OH also causes activity restriction and may cause the complications described [2].

Functional electrical stimulation (FES) is a nonpharmacological intervention for OH following SCI [8]. According to several studies [9,10], FES-induced leg muscle contractions increase cardiac output due to increased venous return. This leads to blood pressure stabilization.

Passive leg movement is also a nonpharmacological intervention for OH following SCI [8]. The reasons have been reported to include stabilization of the decline in central blood volume during orthostatic stress, sympathetic activation by exercise, and increased venous return from lower leg muscle contractions such as FES [9,11,12]. Passive leg movement requires a therapist or equipment such as a step machine and rehabilitation robot. For example, it has been reported that adding passive leg movement using a stepper or ergometer to regular orthostatic training in healthy subjects stabilized blood flow and prevented syncope [13].

We developed a gait rehabilitation robot for people with paraplegia that combined FES with robotic assistance [14] (Figure 1). This robot is a gait rehabilitation robot classified as passive user mode [15], using the stationary system with a robotic exoskeleton and body weight support on a treadmill. The gait trajectory was predetermined by programming the joint angles of a healthy subject [14]. Therefore, regardless of the degree of paralysis, gait training can be performed using passive leg movement. The FES stimuli were also programmed with reference to the gait phase of a healthy subject [14]. This robot can provide active movement with FES and passive movement with robotic assistance like a stepper.

The exoskeleton was originally developed based on the hip–knee–ankle–foot orthosis for paraplegia. The participants were lifted, and their weight was supported with a rehabilitation lift. They walked on a treadmill with functional electrical stimulation.

A robot-assisted tilt table, which can also provide the above two functions (FES and passive movement), has been previously reported to positively affect cardiopulmonary function in post-stroke patients and SCI patients [16,17]. They reported significant increases in the oxygen intake and heart rate during exercise in patients with incomplete SCI [17]. They assessed the oxygen uptake, heart rate, and mean blood pressure during exercise; however, the mean blood pressure was a single measurement, and changes in blood pressure over time during exercise were not measured. The study included only chronic cases (16–46 weeks) and did not focus on the effects of orthostatic stress in the intermediate phase of SCI (2 weeks to 6 months) [17,18]. Therefore, changes in blood pressure over time during exercise owing to the combination of robotic leg movements and FES in the intermediate phase of SCI are unknown.

Recently, it has been reported that robotic rehabilitation combined with FES in upper limb and post-stroke rehabilitation leads to better results than robotics alone [19,20]. However, there are few reports for SCI. In one case series, four SCI patients (3 to 18 months) reported improved walking ability after 5 weeks of robotic training [21]. Therefore, the effect of robotic rehabilitation combined with FES on the acute and intermediate phases of SCI is unknown.

We aimed to investigate the effect of a gait training rehabilitation robot combined with FES on orthostatic stress in intermediate-phase SCI by measuring blood flow and blood pressure over time to address this knowledge gap.

## 2. Materials and Methods

### 2.1. Ethical Considerations

Before recruitment, this study was reviewed and approved by the Ethics Committee of the Certified Clinical Research Review Board, Akita University, and the Ministry of Health, Labor, and Welfare (acceptance no.: A2022-01, jRCTs022220018). All of the participants provided written informed consent before screening and recruitment.

### 2.2. Patient Population

Patients with paraparesis were recruited from our institution between June 2022 and March 2024. Patients with paraparesis were defined as those with SCI due to trauma. The inclusion criteria were as follows: (1) inability to walk unaided; (2) cases with OH during standing; (3) recognition of the significance of this research and participation in this research of their own free will; (4) written informed consent after explanation in the informed consent document. The exclusion criteria were as follows: (1) inability to follow the therapist’s instructions; (2) conditions in which exercise load leads to the deterioration of physical condition; (3) severe joint contractures or deformities; (4) other movement limitations for any reason. OH was defined as a fall in blood pressure upon standing, specifically, a decrease in systolic blood pressure of at least 20 mmHg or a decrease in diastolic blood pressure of at least 10 mmHg within 3 min of standing [22].

### 2.3. Procedure

The training time was set to 10 min for robotic training (RT) and 5 min for tilt table training (TT). RT was set at 10 min based on previous reports [1,23]. TT was set as the minimum training time for patients with acute-phase SCI [24]. If a patient was unable to continue, training was discontinued. Therefore, data from the first 5 min of training for each group were used for comparison. The measurements in this study were taken at the time of the first RT, and the TT was measured as close to the date as possible.

In the RT, we used a gait training rehabilitation robot combined with FES. This robot is composed of an exoskeleton designed based on a hip–knee–ankle–foot orthosis for paraplegia, rehabilitation lift (Moritoh, Aichi, Japan; SP-1000), treadmill (Johnson Health Tech, Tokyo, Japan; 8.1T), and FES (Minato Medical Science, Osaka, Japan; Dynamid, DM2500). This robot has motors on the bilateral knee and hip joints that rotate according to the gait cycle. This allows the robot to provide passive lower limb movements that mimic walking. FES was used on the bilateral quadriceps and hamstrings (Figure 2). The quadriceps, mainly the rectus femoris, were stimulated from the terminal swing to the mid-stance. The hamstrings, mainly on the lateral side, were stimulated from the mid-stance to the pre-swing phase. The stimulus settings were 25 Hz and 20–40 mA, which was set as the minimum stimulus that caused muscle contraction in the target muscle. The area containing the quadriceps muscle fibers was identified and used as the anterior stimulation point by palpating the inferior anterior iliac spine and patella [25]. Similarly, the area containing hamstring muscle fibers was identified and used as the posterior stimulation point by palpating the sciatic tuberosity and the head of the fibula [25]. These systems provided body weight-supported treadmill gait training combined with FES. RT was performed in a fully upright position (90°).

We used a tilt table for TT (Minato Medical Science, Osaka, Japan; tilt table K1430MN). The tilt table angle was set to the maximum angle that could be achieved based on the rehabilitation progress (maximum 80°). This training’s success was defined as the ability to stand up by more than 60° with reference to the head-up tilting table test [26].

### 2.4. Measurement

We measured the earlobe blood flow (EBF) and non-invasive blood pressure (NIBP) to observe hemodynamics during exercise and rest. The EBF was measured using a wireless laser Doppler blood flow meter (JMS, Tokyo, Japan; Pocket LDF) (Figure 3). This device can measure the EBF every second, and stable measurement is possible even during exercise [27]. The relative EBF was calculated by dividing the measured EBF by the mean EBF at rest for 2 min, since the measured EBF was affected by the thickness of the earlobe and the number of capillaries and varies from individual to individual [27]. This device can also measure the pulse rate every second. NIBP was measured every 2 min using a patient monitor (Fukuda Denshi, Tokyo, Japan; DSC-7300). Resting blood pressure measurements were taken in the supine position before training. Each measurement was initiated at the onset of lower limb movement for RT and at the onset of standing for TT. The angle was recorded for TT. The time available for each training session was measured, and success was defined as a training session that lasted for at least 5 min.

Questionnaires were administered after RT to obtain subjective feedback. The questionnaires included the following four items to evaluate positive and negative sensations: “Sense of Achievement”, “Willingness”, “Discomfort”, and “Fatigue”. They were answered on a 5-point scale from 1 (not applicable) to 5 (highly applicable), referring to a previous study [28]. We also evaluated the occurrence of adverse events.

### 2.5. Statistical Analysis

The EBF from the first 5 min of RT and TT was used for comparison. The slope of the regression line was calculated at 0.5 min and every minute to obtain the rate of change in the EBF and pulse rate. The time windows were 0.5–1 min, 1–2 min, 2–3 min, 3–4 min, and 4–5 min.

A Wilcoxon rank-sum test was then used to compare the slopes at RT and TT at each time point. A linear mixed model was analyzed with the slopes as the dependent variable, training session and time as fixed effects, and individual differences as random effects. NIBP was averaged separately during exercise and at rest for each training session. If an error occurred during measurement, the value was assumed to be missing. Systolic blood pressure and mean blood pressure were compared during exercise and at rest for each exercise session and between RT and TT. Paired *t*-tests were used for comparison. Statistical analyses were conducted using EZR (Saitama Medical Center, Jichi Medical University, Saitama, Japan) [29]. Statistical significance was set at *p* = 0.05.

## 3. Results

Among our participants, five males and one female completed both exercise sessions (Table 1). The mean patient age was 49.5 years. Four patients had quadriplegia and two had paraplegia. All of the patients were unable to walk. All of the cases began the intervention within 3 months of injury, with a mean time from injury of 39.3 days. Cases 5 and 6 were SCIs without radiographic abnormality, but the MRI showed signal changes within the spinal cord in the T2-weighted images [30,31]. TT measurements were taken within 15 days of RT.

Figure 4 shows the EBF and systolic blood pressure during the first 5 min of each exercise session. EBF decreased immediately after the start of TT in all cases, whereas it increased after the start of RT in five cases.

The slopes of the regression line at 0.5 min and every 1 min in RT and TT were calculated to clarify this change. The 0.5-min slope in TT was negative, whereas the RT slope was positive, indicating a significant difference between the two groups (*p* = 0.03). The 1 min slope and that after were close to zero, with the following *p*-values: *p* = 0.16 at 1 min; *p* = 0.84 at 2 min; *p* = 0.84 at 3 min; *p* = 0.31 at 4 min; *p* = 0.31 at 5 min (Figure 5). A linear mixed-model analysis showed no variations in individual differences, and the time points did not affect the slope at any time point (*p* > 0.05). The training type had a significant effect on the slope of the EBF (*p* < 0.01) (Table 2).

Additionally, the slope of the regression line was calculated for the pulse rate. The 1-min slope in TT was positive, whereas the RT slope was negative, indicating a significant difference between the two groups (*p* = 0.03). The slope at 0.5 min, 2 min, and after was close to zero, with the following *p*-values: *p* = 0.16 at 0.5 min; *p* = 0.84 at 2 min; *p* = 0.56 at 3 min; *p* = 0.31 at 4 min; *p* = 0.31 at 5 min (Figure 6). A linear mixed-model analysis showed no variation in individual differences (variance = 0; standard deviation = 0), and the time points did not affect the slope at any time point (*p* > 0.05). The training type had a significant effect on the slope of the pulse rate (*p* < 0.05) (Table 3).

Table 4 shows the average systolic and mean blood pressures during exercise and at rest in RT and TT. The differences in systolic blood pressure between exercise and rest were −15.7 (95% confidence interval [CI]: −29.2 to −2.2) for RT (*p* = 0.03) and −22.6 (95% CI: −43.7 to −1.5) for TT (*p* = 0.04). The differences in mean blood pressure between exercise and rest were −13.3 (95% CI: −21.8 to −4.8) for RT (*p* = 0.01) and −18.6 (95% CI: −34.1 to −3.2) for TT (*p* = 0.03). The differences between RT and TT were 6.3 (95% CI: −9.6 to 22.3) for systolic blood pressure (*p* = 0.35) and 4.4 (95% CI: −7.8 to 16.7) for mean blood pressure (*p* = 0.40).

Table 5 presents the questionnaire results. Three patients responded with a score of five on the sense of achievement, while two patients responded with a score of four. Regarding the willingness to continue, three cases responded with a score of five, and three cases responded with a score of four. Four patients had the lowest discomfort score of one.

All of the patients were able to undergo RT for more than 5 min, whereas only four cases were able to undergo TT. Case 1 discontinued treatment owing to dizziness in TT. Case 2 involved the inability to perform TT at 60° and instead performing it at 45°. In contrast, Case 1, who underwent RT, initially experienced dizziness, which gradually decreased during exercise.

## 4. Discussion

In this study, we used a gait training rehabilitation robot combined with FES for robotic training. In the EBF, robotic training was able to maintain the blood flow and pulse rate. In addition, systolic and mean blood pressure were slightly higher in robotic training than in tilt table training, although the differences were not significant. These results suggest that hemodynamic stability may be maintained during robotic training in response to orthostatic stress. As a result, subjective symptoms might be reduced, and the standing time increased in two cases.

We used the EBF for monitoring. Goma developed the EBF as a measurement technique [32]. They reported that the EBF is useful for predicting changes in orthostatic arterial blood pressure because it changes similarly to the mean blood pressure measured using non-invasive tonometry monitoring [32]. It is reported [33] that EBF measurements are used to estimate changes in blood pressure during hemodialysis and [34] that EBF can be used to estimate blood pressure during cardiopulmonary resuscitation. Therefore, we used the EBF to compensate for possible missing values in the non-invasive blood pressure measurements. In this study, subjective symptoms such as dizziness were present in two patients, but there was no difference in blood pressure changes. The changes in the EBF and pulse rate that were measured using this device may have quantified the occurrence of subjective symptoms.

Robotic gait training for patients with SCI has become available for clinical use in recent years and has been reported to be effective because of its many advantages, such as the ability to provide accurate gait trajectories, reduce the human load, and increase the training amount [1]. However, a problem with conventional robotic gait training is that patients can only perform passive movements. Training with muscle contraction independent of the robot’s motor torque may be more useful in the rehabilitation of people with paraplegia, whereas such training is virtually impossible with a robot alone. We focused on FES and developed a gait rehabilitation robot combined with FES [14] to compensate for this disadvantage. A previous report suggested the possibility of muscle strengthening and improved walking ability [21]. The precise gait trajectory provided by the robot may also improve gait patterns [35].

This study examined the immediate effects of robotic training, focusing on its ability to prevent OH in the intermediate phase of SCI. In a previous report, a robot-assisted tilt table capable of performing FES and passive movement of the legs, which are effective for OH in the chronic phase of SCI [8], was found to maintain blood pressure [17]. We found similar results in the intermediate phase of SCI.

Possible factors involved in the development of OH in SCI include sympathetic nervous system dysfunction, altered baroreceptor sensitivity, a lack of skeletal muscle pumps, cardiovascular deconditioning, and an altered salt and water balance [36]. Sympathetic preganglionic neurons that control blood pressure are located between the T1 and L2 spinal segments [37]. This suggests that OH is caused by damage or dysfunction at the spinal cord level. Several interventions exist for OH [8], including pharmacologic and nonpharmacologic approaches; however, the only way to fundamentally solve the problem is to acclimate the patient to standing, such as with a tilt table. RT can provide standing alongside gait training and can be utilized for a longer term. The long-term effects of RT on OH are unknown and should be investigated in future studies. Furthermore, long-term RT is known to provide benefits, such as increased lower-extremity muscle strength and improved gait ability, functional status, and quality of life [38,39]. FES improves spasticity and balance [40], suggesting that the combination of robotics and FES may improve the effectiveness of rehabilitation. There was some concern that the use of FES might affect comfort, but four patients reported no discomfort in the questionnaire (a score of one). In addition, four patients felt fatigued (scores of four and five) despite little change in their blood flow or pulse rate, which may have been due to sufficient lower limb movement with muscle contraction via FES. All cases were motivated for the next training (scores of four and five). The results of this questionnaire show that robotic training can be performed over a longer term. Future studies should further investigate the effects of long-term training.

Our study had some limitations. First, we enrolled only a small number of patients. There were no significant differences in systolic and mean blood pressure values between RT and TT. A power analysis of the data from this study with a blood pressure difference of 10 and a standard deviation of 15 resulted in 20 cases. However, despite the small numbers, the EBF and pulse rate changes differed. A larger sample size is needed to confirm this study’s generalizability. Second, the systolic blood pressure measurement was non-invasive and had to be determined as a missing value if it could not be measured. We used the EBF to compensate for possible missing values in the non-invasive blood pressure measurements. Future studies should also consider using invasive blood pressure measurements to measure the systolic blood pressure continuously and reliably. Third, measurements were taken within 15 days of each training session. Spontaneous recovery during this period could not be considered. This means that, during this period, the patients might have recovered and been able to withstand the OH. However, as TT was performed after RT in five out of six cases, it is possible that the blood pressure changes were more stable in the TT group. Future studies should be conducted under similar conditions; for example, each training session should be conducted within 24 h. Fourth, it was difficult to measure the blood pressure in the supine position at rest during RT. Therefore, the blood pressure before moving to the rehabilitation room was used as a substitute. Future studies need to measure the blood pressure under more appropriate timing and conditions.

## 5. Conclusions

We performed robotic training using a gait training rehabilitation robot combined with FES on patients during the intermediate phase of SCI. Despite orthostatic stress during robotic training, blood flow was maintained during the exercise. Our results suggest that robotic training with FES can stabilize blood pressure changes early in the intermediate phase of SCI and can be safely performed. This study investigated a small number of cases; future studies should examine a larger number of cases and long-term effects.

## Figures and Tables

**Figure 1 sensors-25-00984-f001:**
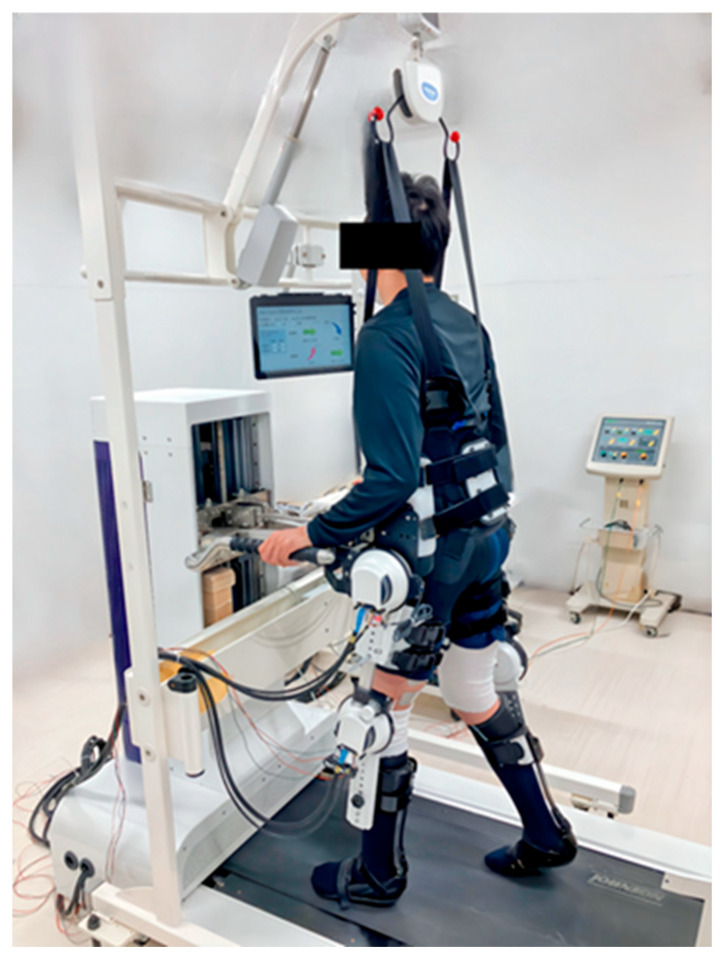
A gait rehabilitation robot combined with functional electrical stimulation: Akita Trainer.

**Figure 2 sensors-25-00984-f002:**
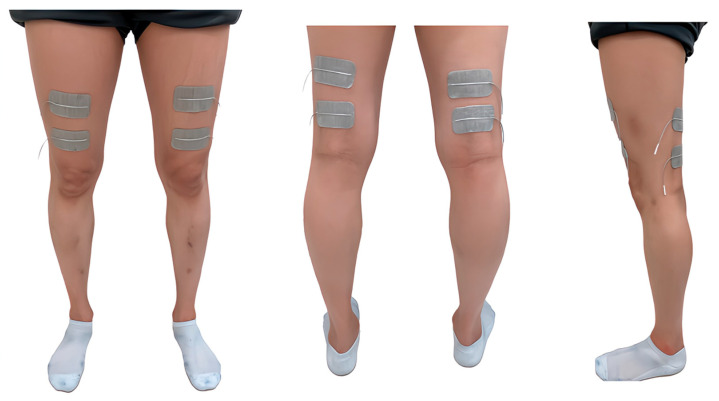
The positions of the functional electrical stimulation pads. The pads were placed on the quadriceps and hamstrings on both sides (silver pads).

**Figure 3 sensors-25-00984-f003:**
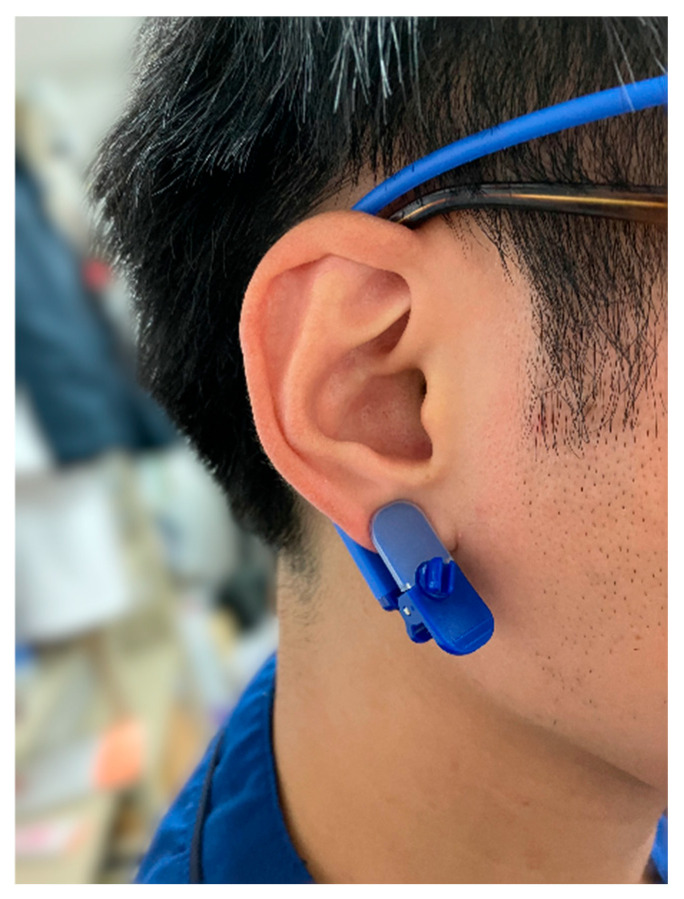
Wireless laser Doppler blood flow meter: Pocket LDF.

**Figure 4 sensors-25-00984-f004:**
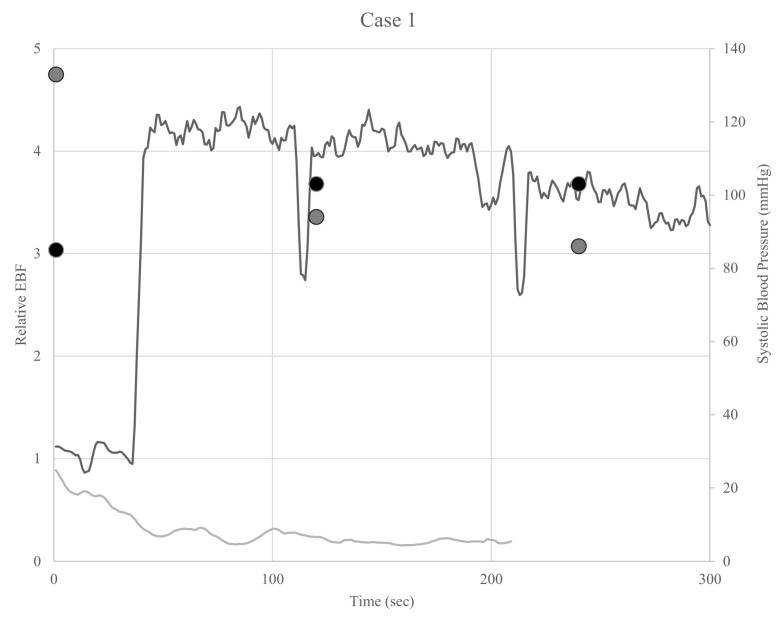
Graphs of the EBF ratio and systolic blood pressure after training. Black and gray lines indicate the EBF ratios in robot training and tilt table training, respectively. Black and gray dots indicate the systolic blood pressure during robot training and tilt table training, respectively.

**Figure 5 sensors-25-00984-f005:**
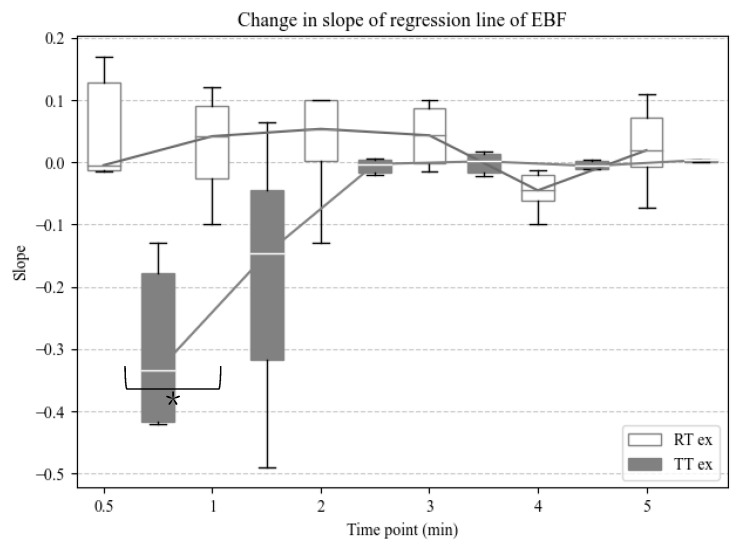
Box and whisker plot showing changes in the slope of the regression line of the EBF at 30 s and every 1 min. White indicates robot training and dark gray indicates tilt table. *: *p* < 0.05 vs. TT exercise group at 0.5 min using the Wilcoxon rank-sum test.

**Figure 6 sensors-25-00984-f006:**
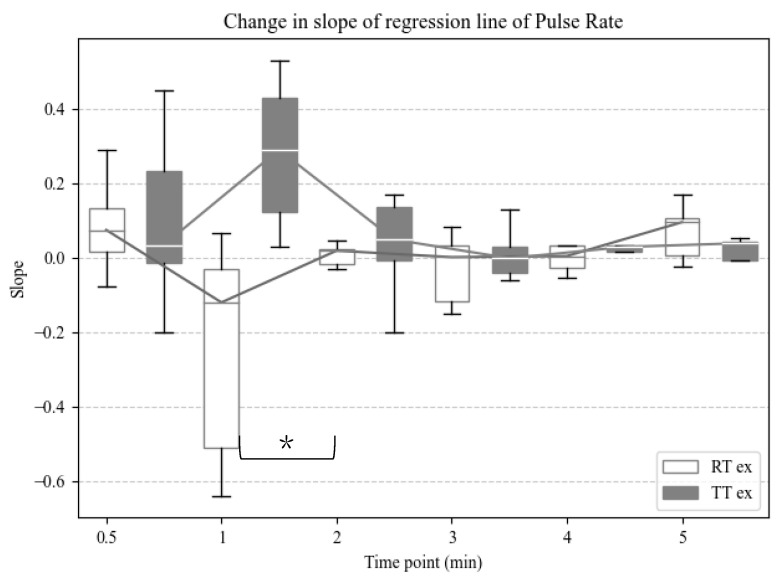
Box and whisker plot showing changes in the slope of the regression line of pulse rate at 30 s and every 1 min. White indicates robot training and dark gray indicates tilt table. *: *p* < 0.05 vs. TT exercise group at 0.5 min using the Wilcoxon rank-sum test.

**Table 1 sensors-25-00984-t001:** Patients’ demographics and clinical characteristics.

Case	Sex	Age (Years)	Cause	Neurological Level of Injury	AIS	Time Post-Injury (Days)
1	M	23	C6 vertebral fracture	C4	B	68
2	M	38	T11–12 dislocation fracture	T10	C	45
3	M	33	L1 vertebral fracture	T11	D	25
4	M	68	C7–T1 dislocation fracture	C7	C	23
5	F	73	SCI without radiographic abnormality	C3	C	24
6	M	62	SCI without radiographic abnormality	C4	C	51
Average		49.5				39.3

AIS: American Spinal Injury Association Impairment Scale. SCI: spinal cord injury.

**Table 2 sensors-25-00984-t002:** Linear mixed model for the slope of the regression line of the EBF from 0.5 min for 5 min per training session. *: *p* < 0.05.

Fixed Effects	Estimate	Standard Error	*p*-Value
Intercept	−0.011	0.058	0.85
Training	−0.14	0.04	0.0028 *
Time: 0.5 min	−0.081	0.075	0.28
Time: 2 min	−0.11	0.075	0.13
Time: 3 min	0.092	0.075	0.22
Time: 4 min	0.039	0.077	0.61
Time: 5 min	0.088	0.077	0.26
Random Effects	Group	Variance	
Case	Intercept	0.00049 ± 0.022	
Residual		0.034 ± 0.18	

**Table 3 sensors-25-00984-t003:** Linear mixed model for the slope of the regression line of the pulse rate from 0.5 min for 5 min per training session. *: *p* < 0.05.

Fixed Effects	Estimate	Standard Error	*p*-Value
Intercept	−0.24	0.12	0.060
Training	0.20	0.095	0.042 *
Time: 0.5 min	0.24	0.16	0.15
Time: 2 min	0.15	0.16	0.34
Time: 3 min	0.028	0.16	0.86
Time: 4 min	0.20	0.17	0.24
Time: 5 min	0.13	0.17	0.43
Random Effects	Group	Variance	
Case	Intercept	0.00 ± 0.00	
Residual		0.16 ± 0.40	

**Table 4 sensors-25-00984-t004:** Table of systolic blood pressure and mean blood pressure during exercise and at rest between the robot and tilt table training.

RT	Exercise	Rest	Difference (95% CI)	*p*
Systolic BP	99.1 ± 9.8	114.8 ± 9.8	−15.7 (−29.2 to −2.2)	0.03
Mean BP	71.9 ± 6.9	85.2 ± 5.7	−13.3 (−21.8 to −4.8)	0.01
TT	Exercise	Rest	Difference (95% CI)	*p*
Systolic BP	92.8 ± 14.8	115.4 ± 14.0	−22.6 (−43.7 to −1.5)	0.04
Mean BP	67.4 ± 11.8	86.1 ± 11.6	−18.6 (−34.1 to −3.2)	0.03
	RT	TT	Difference (95% CI)	*p*
Systolic BP	99.1 ± 9.8	92.8 ± 14.8	6.3 (−9.6 to 22.3)	0.35
Mean BP	71.9 ± 6.9	67.4 ± 11.8	4.4 (−7.8 to 16.7)	0.40

RT: robot training. TT: tilt table training. BP: blood pressure. CI: confidence interval.

**Table 5 sensors-25-00984-t005:** The results of the post-robot training questionnaires.

	Sense of Achievement	Fatigue	Discomfort	Willingness
Case 1	3	2	1	5
Case 2	4	4	3	4
Case 3	4	5	2	4
Case 4	5	4	1	4
Case 5	5	3	1	5
Case 6	5	5	1	5
Average	4.3	3.8	1.5	4.5

Range: 1–5 (1: not applicable; 5: applicable).

## Data Availability

The data are contained within this article. Please contact the authors for additional data.

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
