# Peer review of "Effects of Gait Rehabilitation Robot Combined with Electrical Stimulation on Spinal Cord Injury Patients’ Blood Pressure"

_sensors, 2025, doi:10.3390/s25030984_

Round 1

Reviewer 1 Report

Comments and Suggestions for Authors

attached

Comments on the Quality of English Language

attached

Author Response

Thank you for your comments on our manuscript. We have responded to your comments and included them in red.

  1. Sample Size and Statistical Power:
  • Weakness: The study only includes six participants, which limits generalizability and statistical significance.
  • Recommendation: Address the sample size limitation in greater depth, possibly using a power analysis to justify or recommend the sample size for future studies. Consider discussing the feasibility of expanding the study.

→The results of the power analysis using this study have been added to the limitation(Line 363-366).

  1. Results Interpretation:
  • Weakness: While differences in EBF and pulse rate are reported as significant, the clinical relevance of these changes is not well-elaborated.
  • Recommendation: Include a discussion on the implications of EBF and pulse rate changes for patient outcomes and rehabilitation practices.

→ We have added a note in the discussion section that the subjective symptoms had improved in the two cases (Line 311-312, 320-323).

  1. Limitations Section:
  • Weakness: While the authors acknowledge limitations, some are inadequately addressed, such as the potential influence of spontaneous recovery or differences in measurement timing.
  • Recommendation: Expand on how these limitations might affect results and propose more robust designs to mitigate them in future research.

→ We have added the sentence “ This means that during this period, the patients might have recovered and been able to withstand the OH.” to Limitation 3. (Line 372-373).

  1. Questionnaire Data:
  • Weakness: Patient-reported outcomes, while valuable, are subjective and not correlated with physiological measures.
  • Recommendation: Provide a deeper analysis of questionnaire responses, possibly correlating them with physiological data to strengthen conclusions about patient experience.

→We have added a discussion of the questionnaire and the actual measurement data in this study. (Line 354-359).

  1. Statistical Analysis:
  • Weakness: Some statistical tests, such as the Wilcoxon rank-sum test, may not account for within-subject correlations in this repeated-measures design.
  • Recommendation: Reassess the statistical approach to confirm the robustness of the findings. Consider mixed-effects models to account for within-subject variability.

→Thank you for pointing this out. We have added a linear mixed model and added the method and results (Line 187-189, 239-242, 258-261, Table 2, Table 3).

  1. Writing Style:
  • Weakness: There are grammatical inconsistencies, such as "The in Spinal cord injury" in the Introduction.
  • Recommendation: Perform a thorough language and grammar check. Consider professional editing to improve clarity and readability.

→We requested an English editing service.

Specific Comments:

Abstract:

  • The abstract summarizes the study well but should briefly mention the limitations to set realistic expectations for readers.

→We added “Limitation” to the abstract and adjusted the number of characters.

Introduction:

  • The introduction provides a strong rationale but can benefit from more recent references and a concise description of the knowledge gap addressed by the study.

→We have made revisions, including the reference materials you provided.

Methods:

  • Ethical considerations are well-covered, but more detail on the calibration and reliability of devices like the laser Doppler flowmeter would enhance the section.

→We have added an explanation of the device (Line 162).

 Discussion:

  • The discussion successfully ties findings to prior research but misses opportunities to explore broader implications of robotic interventions in SCI rehabilitation.

→We mentioned in the discussion that it is possible to train with an accurate walking trajectory and that it is possible to reduce the human load and increase the amount of training (Line 325-327). Regarding reducing medical costs, we thought that there would be differences in the introduction costs etc. depending on the country, so we did not add this to the paper.

 Conclusion:

  • While concise, the conclusion could explicitly emphasize how the study lays ground work for larger-scale trials or integration into clinical practice.

→We added a limitation to the Conclusion.

References:

  • The following recent related publication is highly recommended to be refeercned:

https://ieeexplore.ieee.org/document/10286847

→We added the article to the citations and revised the main text (Line 66).

Reviewer 2 Report

Comments and Suggestions for Authors

Minor Revisions:

1. Figure 2: The figure appears incomplete. The title lacks specificity, and it would be beneficial to highlight the positions of the functional electrical stimulation pads in the figure. Additionally, arranging the three images in a 1x3 layout would be sufficient.

2. Figure 3: Proofreading of the figure title is required.

3. Figures (except Figure 3): The resolution of all other figures needs improvement.

Major Comments:

1. Proofreading of the title is necessary. For instance, a more appropriate title could be: Effects of Gait Rehabilitation Robot Combined with Electrical Stimulation on Spinal Cord Injury Patients’ Blood Pressure. The current title is ambiguous and unclear.

2. The literature review in the introduction is insufficient, and the paragraph structure (e.g., at the top of page 2) is poorly organized. This section needs significant revision to improve clarity and coherence.

3. The figure 1 requires detailed labeling and descriptions of the key components of the gait rehabilitation robot. The current title also lacks specificity and should be revised.

4. The rationale for investigating the changes in blood pressure characteristics during lower limb rehabilitation is weak. The logical connections and justification for the study’s importance need to be significantly strengthened.

Comments on the Quality of English Language

The English writing quality of this paper is highly inadequate. It must undergo thorough proofreading and editing before resubmission.

Author Response

Thank you for your comments on our manuscript. We have responded to your comments and included them in red.

  1. Figure 2: The figure appears incomplete. The title lacks specificity, and it would be beneficial to highlight the positions of the functional electrical stimulation pads in the figure. Additionally, arranging the three images in a 1x3 layout would be sufficient.

→As you pointed out, we have corrected Figure 2.e corrected it as you pointed out.

  1. Figure 3: Proofreading of the figure title is required.

→The title of Figure 3 has been corrected to “Wireless laser Doppler blood flow meter: Pocket LDF”.

  1. Figures (except Figure 3): The resolution of all other figures needs improvement.

→As you pointed out, we have adjusted the resolution.

Major Comments:

  1. Proofreading of the title is necessary. For instance, a more appropriate title could be: Effects of Gait Rehabilitation Robot Combined with Electrical Stimulation on Spinal Cord Injury Patients’ Blood Pressure. The current title is ambiguous and unclear.

→We corrected the title as you pointed out.

  1. The literature review in the introduction is insufficient, and the paragraph structure (e.g., at the top of page 2) is poorly organized. This section needs significant revision to improve clarity and coherence.

→We have reconsidered and revised the content of the Introduction.

  1. Figure 1 requires detailed labeling and descriptions of the key components of the gait rehabilitation robot. The current title also lacks specificity and should be revised.

→We have corrected the legend for Figure 1.

  1. The rationale for investigating the changes in blood pressure characteristics during lower limb rehabilitation is weak. The logical connections and justification for the study’s importance need to be significantly strengthened.

→We have added the details about OH  to the Introduction as you pointed out (Line 45-49).

The English writing quality of this paper is highly inadequate. It must undergo thorough proofreading and editing before resubmission.

→We requested an English editing service.

Reviewer 3 Report

Comments and Suggestions for Authors

    This work investigated whether a gait rehabilitation robot combined with functional electrical stimulation (FES) stabilizes blood pressure during orthostatic stress in SCI. A few suggestions are as follows:

1. The expected study in this paper (2-24weeks) overlaps with the existing study (16-46weeks), and the study data in this paper were obtained from patients with onset of disease around 3-10weeks. What is the more significant medical basis for this time range? Do the 3-10weeks data represent the 2-24weeks of the expected study?

2. The definition of passive movement is not clear, As in "passive movement of the lower extremities has been reported to increase venous return through a muscle pump effect like FES "and" This robot can provide active movement with FES and passive movement with robot-assisted locomotion for paralyzed lower limbs. "This is an ambiguous description. In the field of exoskeleton robotics, there are also relevant definitions of active motion and passive motion, which should be further distinguished and more rigorous descriptions should be adopted.

3. The number of subjects and samples are relatively small, human blood pressure is affected by many factors, and the results are derived from the comparison of blood pressure separated by several days and the changes measured within a short time. Can more analytical or medical evidence be provided to support the conclusion?

Author Response

Thank you for your comments on our manuscript. We have responded to your comments and included them in red.

  1. The expected study in this paper (2-24weeks) overlaps with the existing study (16-46weeks), and the study data in this paper were obtained from patients with onset of disease around 3-10weeks. What is the more significant medical basis for this time range? Do the 3-10weeks data represent the 2-24weeks of the expected study?

→We wanted to verify the effects of the acute phase the most, but due to the course of the patient's general condition, it ended up being 3-10 weeks. This is a different period from the chronic phase, so we used the intermediate phase of SCI (2 weeks to 6 months) as defined.

  1. The definition of passive movement is not clear, As in "passive movement of the lower extremities has been reported to increase venous return through a muscle pump effect like FES "and" This robot can provide active movement with FES and passive movement with robot-assisted locomotion for paralyzed lower limbs. "This is an ambiguous description. In the field of exoskeleton robotics, there are also relevant definitions of active motion and passive motion, which should be further distinguished and more rigorous descriptions should be adopted.

→We have added details to the main text to clarify that our robot is in passive mode and that the content is about passive movement (Line 65-69, 69-72).

  1. The number of subjects and samples are relatively small, human blood pressure is affected by many factors, and the results are derived from the comparison of blood pressure separated by several days and the changes measured within a short time. Can more analytical or medical evidence be provided to support the conclusion?

→I added a linear mixed model analysis and corrected limitation (Line 361-366).

Round 2

Reviewer 1 Report

Comments and Suggestions for Authors

The manuscript has been imporved significantly and it is highly recommended for publication at the current version.